# Magnesium and Calcium Transport along the Male Rat Kidney: Effect of Diuretics

**Pritha Dutta [1,]\*** and **Anita T. Layton [1,2,3,4]**

1    Department of Applied Mathematics, University of Waterloo, Waterloo, ON N2L 3G1, Canada; anita.layton@uwaterloo.ca
2    Cheriton School of Computer Science, University of Waterloo, Waterloo, ON N2L 3G1, Canada
3    Department of Biology, University of Waterloo, Waterloo, ON N2L 3G1, Canada
4    School of Pharmacy, University of Waterloo, Waterloo, ON N2L 3G1, Canada
\*    Correspondence: p7dutta@uwaterloo.ca

**Abstract:** Calcium ($Ca^{2+}$) and magnesium ($Mg^{2+}$) are essential for cellular function. The kidneys play an important role in maintaining the homeostasis of these cations. Their reabsorption along the nephron is dependent on distinct trans- and paracellular pathways and is coupled to the transport of other electrolytes. Notably, sodium ($Na^+$) transport establishes an electrochemical gradient to drive $Ca^{2+}$ and $Mg^{2+}$ reabsorption. Consequently, alterations in renal $Na^+$ handling, under pathophysiological conditions or pharmacological manipulations, can have major effects on $Ca^{2+}$ and $Mg^{2+}$ transport. One such condition is the administration of diuretics, which are used to treat a large range of clinical conditions, but most commonly for the management of blood pressure and fluid balance. While the pharmacological targets of diuretics typically directly mediate $Na^+$ transport, they also indirectly affect renal $Ca^{2+}$ and $Mg^{2+}$ handling through alterations in the electrochemical gradient. To investigate renal $Ca^{2+}$ and $Mg^2$ handling and how those processes are affected by diuretic treatment, we have developed computational models of electrolyte transport along the nephrons. Model simulations indicate that along the proximal tubule and thick ascending limb, the transport of $Ca^{2+}$ and $Mg^{2+}$ occurs in parallel with $Na^+$, but those processes are dissociated along the distal convoluted tubule. We also simulated the effects of acute administration of loop, thiazide, and K-sparing diuretics. The model predicted significantly increased $Ca^{2+}$ and $Mg^{2+}$ excretions and significantly decreased $Ca^{2+}$ and $Mg^{2+}$ excretions on treatment with loop and K-sparing diuretics, respectively. Treatment with thiazide diuretics significantly decreased $Ca^{2+}$ excretion, but there was no significant alteration in $Mg^{2+}$ excretion. The present models can be used to conduct in silico studies on how the kidney adapts to alterations in $Ca^{2+}$ and $Mg^{2+}$ homeostasis during various physiological and pathophysiological conditions, such as pregnancy, diabetes, and chronic kidney disease.

**Keywords:** calcium homeostasis; magnesium homeostasis; electrolyte transport; kidney; renal transport

## 1. Introduction

The divalent cations, $Ca^{2+}$ and $Mg^{2+}$, are important for various physiological processes. About 99% of the body's $Ca^{2+}$ is stored in bones, where it forms a calcium-phosphate compound called hydroxyapatite [1]. The remaining 1% of body calcium plays an important role in various other physiological processes, such as cell signaling, both skeletal and smooth muscle contraction, and blood clotting [1]. $Mg^{2+}$ plays a pivotal role in energy-demanding metabolic reactions, protein synthesis, ensuring membrane integrity, facilitating nervous tissue conduction, promoting neuromuscular excitability, regulating muscle contraction, influencing hormone secretion, and participating in intermediary metabolism. Nearly 99% of the body's $Mg^{2+}$ is distributed within cells or stored in bone, with only a small fraction in circulation [2]. Tight regulation of the serum $Ca^{2+}$ and $Mg^{2+}$ concentrations is essential since too much or too little $Ca^{2+}$ or $Mg^{2+}$ can have dangerous, potentially fatal consequences. To maintain $Ca^{2+}$ and $Mg^{2+}$ balance, it is crucial to regulate the fluxes of $Ca^{2+}$

and $Mg^{2+}$ among the primary organs involved in their regulation, namely the intestine, bone, and kidneys.

The kidneys play an important role in maintaining $Mg^{2+}$ and $Ca^{2+}$ homeostasis. The majority, ~60–70% of the filtered $Ca^{2+}$, is reabsorbed along the proximal tubule through the paracellular pathway [3]. By contrast, paracellular $Mg^{2+}$ permeability in the proximal tubule is very low, and hence only 15–25% of the filtered $Mg^{2+}$ is reabsorbed along this segment [3]. The majority of the filtered $Mg^{2+}$ is reabsorbed along the cortical thick ascending limb (60–70%) paracellularly [3]. The paracellular fractional reabsorption of $Ca^{2+}$ along the thick ascending limb is ~15–25% [3]. The distal convoluted tubule is the final segment that reabsorbs $Mg^{2+}$; hence, it plays an important role in fine-tuning urinary $Mg^{2+}$ excretion. Approximately 5–10% of the filtered $Mg^{2+}$ is reabsorbed transcellularly along the distal convoluted tubule, mediated by the transient receptor potential melastatin 6/7 (TRPM6/7) heteromeric complex on the apical membrane and the $Na^+/Mg^{2+}$ exchanger on the basolateral membrane [3]. Approximately 5–10% of the filtered $Ca^{2+}$ is reabsorbed transcellularly along the distal convoluted tubule and connecting tubule, mediated by the transient receptor potential vanilloid 5 (TRPV5) on the apical membrane, the $Na^+/Ca^{2+}$ exchanger (NCX1), and plasma membrane $Ca^{2+}$-ATPase (PMCA) on the basolateral membrane [3]. Finally, ~2–5% of the filtered $Mg^{2+}$ and $Ca^{2+}$ are excreted through urine [3].

Our understanding of $Ca^{2+}$ and $Mg^{2+}$ handling within different segments of the nephron has been greatly advanced through micropuncture and microperfusion studies in rodent nephrons [4,5]. Furthermore, recent genetic studies have expanded our knowledge about the protein mediators of $Ca^{2+}$ and $Mg^{2+}$ transport [6]. Despite these advances, our understanding of the renal handling of these electrolytes remains incomplete. What fraction of the renal reabsorption goes through the transcellular versus paracellular pathway? To what extent is the renal transport in each nephron segment coupled to the transport of other electrolytes, e.g., $Na^+$, $K^+$, and $Cl^-$? To answer these questions, we developed a detailed computational model of epithelial transport of electrolytes and water along the nephrons in a male rat kidney and conducted simulations to predict the renal transport of $Ca^{2+}$ and $Mg^{2+}$ as well as other electrolytes and water under different physiological conditions.

Besides electrolyte and fluid homeostasis, the kidney also plays an essential role in maintaining normal blood pressure. For the management of blood pressure and fluid balance, diuretics are commonly prescribed. Although the pharmacological targets of diuretics directly affect $Na^+$ transport, they also indirectly affect renal $Mg^{2+}$ and $Ca^{2+}$ reabsorption through changes in the electrochemical gradient. How is renal $Ca^{2+}$ and $Mg^{2+}$ transport affected by the administration of diuretics? To answer this question, we simulate the effect of acute administration of three classes of diuretics—loop; thiazide; and K-sparing diuretics—on renal $Mg^{2+}$ and $Ca^{2+}$ transport and excretion.

## 2. Materials and Methods

We have previously developed epithelial cell-based computational models of transporter-mediated solute and water transport along the nephron of a rat kidney [7–10], focusing on the renal handling of $Na^+$, $K^+$, $Ca^{2+}$, glucose, and water in physiological and pathophysiological conditions. The superficial nephron model includes the proximal tubule, short descending limb, thick ascending limb, distal convoluted tubule, connecting tubule, and collecting duct segments. Each nephron segment is represented as a tubule lined by a layer of epithelial cells. The model tracks the transport of the following 17 solutes: $Na^+$, $K^+$, $Cl^-$, $HCO_3^-$, $H_2CO_3$, $CO_2$, $NH_3$, $NH_4^+$, $HPO_4^{2-}$, $H_2PO_4^-$, $H^+$, $HCO_2^-$, $H_2CO_2$, urea, glucose, $Ca^{2+}$, and $Mg^{2+}$. The segment and cell type determine the type and abundance of transporters found on the apical and basolateral membranes of the cell. Solutes and water may be transported across the epithelium by either moving across the apical and basolateral membranes in the transcellular pathway, mediated by specialized membrane transporters or channels, or via the paracellular pathway between neighboring cells. A schematic diagram for the model nephrons is shown in Figure 1.

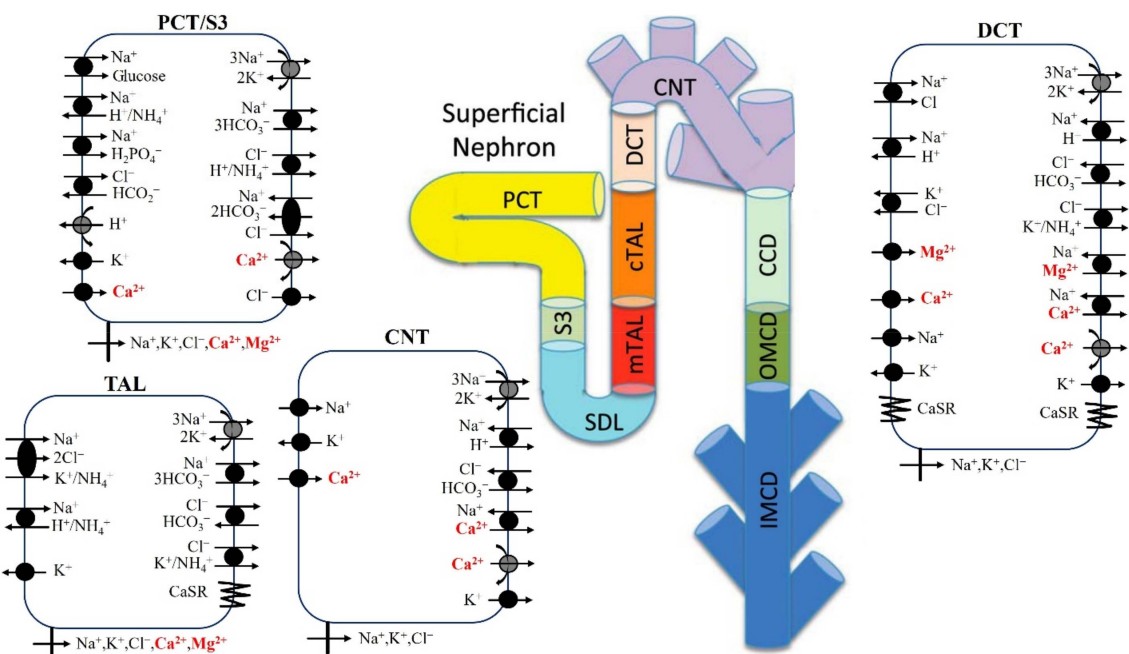

**Figure 1. Model diagram of epithelial transport of Ca²⁺ and Mg²⁺ and selected electrolytes along the superficial nephron.** Mg²⁺ transport occurs along the proximal tubule (proximal convoluted tubule, PCT, and S3), cortical thick ascending limb (cTAL), and distal convoluted tubule (DCT). Ca²⁺ transport occurs along the proximal tubule (proximal convoluted tubule, PCT, and S3), medullary/cortical thick ascending limb (mTAL/cTAL), distal convoluted tubule (DCT), and connecting tubule (CNT). Only the major Na⁺, K⁺, Cl⁻, Ca²⁺, and Mg²⁺ transporters are shown. PCT, proximal convoluted tubule; SDL, short descending limb; mTAL, medullary thick ascending limb; limb; CNT, connecting tubule; CCD, cortical collecting duct; OMCD, outer-medullary collecting duct; IMCD, inner-medullary collecting duct.

The model is defined by a large system of coupled differential and algebraic equations that describe mass conservation and determine transmembrane and paracellular fluxes [11]. The model predicts luminal fluid flow, hydrostatic pressure, membrane potential, luminal and cytosolic solute concentrations, transcellular and paracellular fluxes, urine volume, and urinary excretion rates of model solutes.

Below, we summarize a model representation of Mg²⁺ transport along the proximal tubule, cortical thick ascending limb, and distal convoluted tubule. Model parameters that describe Mg²⁺ transport are given in Table 1. The analogous model description and parameter for Ca²⁺ transport can be found in Ref. [10]. Additional model parameters can be found in Ref. [12].

### 2.1. Mg²⁺ Transport along the Proximal Tubule

The proximal tubule reabsorbs 15–25% of the filtered Mg²⁺ load through the paracellular pathway, which is mediated by claudin-2 and -12 [13] and is driven by the favorable electrochemical gradient established by Na⁺/H⁺ exchanger 3 (NHE3)-mediated Na⁺ reabsorption [5,14,15]. Paracellular electro-diffusive Mg²⁺ flux ($J_{Mg}^{PT,LI}$) is given by

$$J_{Mg}^{PT,LI} = P_{Mg}^{PT,LI} \zeta_{Mg}^{LI} \left( \frac{C_{Mg}^{L} - C_{Mg}^{I} e^{-\zeta_{Mg}^{LI}}}{1 - e^{-\zeta_{Mg}^{LI}}} \right), \tag{1}$$

where the superscripts L and I denote lumen and lateral intercellular space (LIS), respectively; $P_{Mg}^{PT,LI}$ denotes the permeability of Mg²⁺ at the lumen and LIS interface; $\zeta_{Mg}^{LI} = \frac{Z_{Mg}F}{RT}(\psi^{L} - \psi^{I})$; $Z_{Mg}$ is the valence of Mg²⁺ (+2); $C_{Mg}^{L}$ and $C_{Mg}^{I}$ denote Mg²⁺ concentrations in the lumen and LIS, respectively; $\psi^{L}$ and $\psi^{I}$ denote the luminal and LIS membrane potentials, respectively; $RT = 2.57$ J/mmol; and $F = 96.5$ C/mmol represents Faraday's constant.

### 2.2. $Mg^{2+}$ Transport along the Thick Ascending Limb

The medullary thick ascending limb has negligible $Mg^{2+}$ reabsorption [15–17]. In contrast, the cortical thick ascending limb reabsorbs 60–70% of the filtered $Mg^{2+}$ via the paracellular route. That flux is mediated by claudins 16 and 19 [18], and is driven by the electrochemical gradient established by $Na^+$-$K^+$-$Cl^-$ cotransporter 2 (NKCC2)-mediated $Na^+$ transport [9,14]. Paracellular $Mg^{2+}$ transport in the cortical thick ascending limb $J_{Mg}^{cTAL,LI}$ is represented by an expression analogous to Equation (1), with the superscript "PT" replaced by "cTAL".

### 2.3. $Mg^{2+}$ Transport along the Distal Convoluted Tubule

Unlike the proximal tubule and cortical thick ascending limb, where $Mg^{2+}$ transport proceeds passively via the paracelluar route, the reabsorptive process along the distal convoluted tubule is active and transcellular and is mediated by the TRPM6 and TRPM7 (TRPM6/7) heteromeric channels, expressed on the apical membranes [19]. $Mg^{2+}$ flux through TRPM6/7 is given by

$$J_{Mg}^{TRPM6/7} \;=\; N_{TRPM6/7} \times f_{Mg} \times f_{pH} \times \frac{\Delta\psi^{LC} - E_{Mg}^{LC}}{2F}, \tag{2}$$

$f_{Mg}$ and $f_{pH}$ describe the effects of intracellular $Mg^{2+}$ concentration ($C_{Mg}^{C}$) and extracellular pH on TRPM6/7 [20], and are given by

$$f_{Mg} \;=\; \frac{1}{1 + \left(\dfrac{C_{Mg}^{C}}{0.51}\right)^{2}}. \tag{3}$$

$$f_{pH} \;=\; g_{pH7.4}\left(2 - \frac{1}{1 + \dfrac{7.4 - pH^{L}}{7.4 - pH_{1/2}}}\right), \tag{4}$$

where $g_{pH7.4}$ denotes the single channel conductance of TRPM6/7 channel at pH 7.4, $pH^{L}$ denotes the luminal fluid pH, and $pH_{1/2}$ denotes the luminal fluid pH for half-maximal conductance.

$Mg^{2+}$ efflux through the basolateral membrane is assumed to be mediated by an $Na^+$/$Mg^{2+}$ exchanger [19,21], given by

$$J_{Mg}^{NaMgX} \;=\; J_{Mg}^{NaMgX,max} \times f_{NaS} \times f_{NaC} \times f_{MgS} \times f_{MgC}, \tag{5}$$

where $J_{Mg}^{NaMgX,max}$ denotes the maximum $Mg^{2+}$ flux through the $Na^+$/$Mg^{2+}$ exchanger, and the $f$ terms represent regulation by extracellular $Na^+$, $f_{NaS} = \dfrac{\left(c_{Na}^{S}\right)^{2}}{\left(\left(c_{Na}^{S}\right)^{2} + \left(K_{M,NaS}\right)^{2}\right)}$, intracellular $Na^+$, $f_{NaC} = \dfrac{K_{M,NaC}}{\left(c_{Na}^{C} + K_{M,NaC}\right)}$, extracellular $Mg^{2+}$, $f_{MgS} = \dfrac{K_{M,MgS}}{\left(c_{Mg}^{S} + K_{M,MgS}\right)}$, and intracellular $Mg^{2+}$, $f_{MgC} = \dfrac{\left(c_{Mg}^{C}\right)^{2}}{\left(\left(c_{Mg}^{C}\right)^{2} + \left(K_{M,MgC}\right)^{2}\right)}$. In these expressions, $K_{M,NaS}$, $K_{M,NaC}$, $K_{M,MgS}$, and $K_{M,MgC}$ denote the Michaelis-Menten constants.

### 2.4. Calcium-Sensing Receptor

The calcium-sensing receptor (CaSR) regulates not only $Ca^{2+}$ and $Mg^{2+}$ reabsorption in the kidneys but other electrolytes and water as well by modifying transporter activities. $Ca^{2+}$ is the primary ligand for activating CaSR. At equimolar concentrations, $Mg^{2+}$ is 1/2 to 2/3 as potent as $Ca^{2+}$ in activating CaSR [22,23]. We model the effect of CaSR on a given parameter $v$ ($v$ may denote paracellular permeability, NKCC2 activity, ROMK activity, or NCC activity; see below) with the following expression:

$$v \;=\; v^{*}\left(1 + \alpha_{v,Ca}\left(\frac{\left(C_{Ca}^{i}\right)^{4}}{\left(C_{Ca}^{i}\right)^{4} + \left(EC_{50,Ca}\right)^{4}}\right)\right)\left(1 + \alpha_{v,Mg}\left(\frac{\left(C_{Mg}^{i}\right)^{4}}{\left(C_{Mg}^{i}\right)^{4} + \left(EC_{50,Mg}\right)^{4}}\right)\right), \tag{6}$$

where $v^*$ is the value of $v$ in the absence of the effect of CaSR, $c_{Ca}^i$ and $c_{Mg}^i$ denote the concentration of $Ca^{2+}$ and $Mg^{2+}$ in the luminal ($i$ = L) or interstitial ($i$ = S) fluid, and $EC_{50,Ca}$ = 1.25 mM and $EC_{50,Mg}$ = 2.5 mM represent the half-maximal concentrations for $Ca^{2+}$ and $Mg^{2+}$ [24], respectively. CaSR is ubiquitously expressed in the kidney both along the apical and basolateral membranes, with its highest expression being at the basolateral membrane of the cortical thick ascending limb [25]. Hence, we represent $v$ for (i) paracellular permeability, NKCC2 activity, and renal outer-medullary potassium channel (ROMK) in the thick ascending limb, (ii) $Na^+$-$Cl^-$ cotransporter (NCC) activity in the distal convoluted tubule, (iii) $H^+$-ATPase flux in outer-medullary collecting duct type A cells, and (iv) water permeability in the inner-medullary collecting duct. The parameters $\alpha_{v,Ca}$ and $\alpha_{v,Mg}$ are negative if CaSR has an inhibitory effect on $v$ and positive otherwise. Since the effect of $Mg^{2+}$ on CaSR activation is ~50–66% of that of $Ca^{2+}$, we set $\alpha_{v,Mg}$= $0.6\alpha_{v,Ca}$. The values for $\alpha_{v,Ca}$ and $\alpha_{v,Mg}$ for each of the segments are given in Table 1.

**Table 1.** $Mg^{2+}$-specific parameters for all the segments along the superficial nephron. Values marked (*) are adjusted. PT, proximal tubule; TAL, thick ascending limb; DCT, distal convoluted tubule; CD, collecting duct; OMCD, outer-medullary collecting duct; IMCD, inner-medullary collecting duct.

| Parameter | Value |
| --- | --- |
| **PT** | |
| Tight junction permeability to $Mg^{2+}$ at the lumen-LIS interface ($P_{Mg}^{PT,LI}$) | $1.1 \times 10^{-5}$ cm/s [26] |
| Reflection coefficient of tight junction to $Mg^{2+}$ | 0.89 (*) |
| **cTAL** | |
| Tight junction permeability at the lumen-LIS interface in the absence of $Mg^{2+}$ ($P_{Mg}^{TAL,LI*}$) | $38 \times 10^{-5}$ cm/s (*) |
| Maximum half concentration of $Ca^{2+}$ ($EC_{50,Ca}$) | 1.25 mM [24] |
| Maximum half concentration of $Mg^{2+}$ ($EC_{50,Mg}$) | 2.5 mM [24] |
| Hill function coefficient, n | 4 [24] |
| Inhibitory coefficient of $Ca^{2+}$ on tight junction permeability ($\alpha_{P,Ca}$) | $-4/7$ [27] |
| Inhibitory coefficient of $Mg^{2+}$ on tight junction permeability ($\alpha_{P,Mg}$) | $-0.34$ (*) |
| Inhibitory coefficient of $Ca^{2+}$ on NKCC2 activity ($\alpha_{NKCC2,Ca}$) | $-0.4$ [28] |
| Inhibitory coefficient of $Mg^{2+}$ on NKCC2 activity ($\alpha_{NKCC2,Mg}$) | $-0.24$ (*) |
| Inhibitory coefficient of $Ca^{2+}$ on ROMK activity ($\alpha_{ROMK,Ca}$) | $-0.8$ [29,30] |
| Inhibitory coefficient of $Mg^{2+}$ on ROMK activity ($\alpha_{ROMK,Mg}$) | $-0.48$ (*) |
| **DCT** | |
| TRPM6/7 channel density ($N_{TRPM6/7}$) | $26 \times 10^4$ cm$^{-2}$ (*) |
| Single channel conductance of TRPM6/7 at pH 7.4 ($g_{pH7.4}$) | 56.6 pS [20] |
| Luminal pH for half-maximal conductance of TRPM6/7 (pH$_{1/2}$) | 5.5 [20] |
| Maximum $Mg^{2+}$ flux through $Na^+$/$Mg^{2+}$ exchanger ($J_{Mg}^{NaMgX,max}$) | $8.6 \times 10^{-9}$ mmol/cm$^2$/s (*) |
| Intracellular $Mg^{2+}$ half-saturation constant ($K_{M,Mg C}$) | 3.59 M [19,21] |
| Extracellular $Mg^{2+}$ half-saturation constant ($K_{M,Mg S}$) | 1.3 mM [19,21] |
| Intracellular $Na^+$ half-saturation constant ($K_{M,Na C}$) | 12.29 mM [19,21] |
| Extracellular $Na^+$ half-saturation constant ($K_{M,Na S}$) | 87.5 mM [19,21] |
| Excitatory coefficient of $Ca^{2+}$ on NCC activity ($\alpha_{NCC,Ca}$) | 0.5 [31] |
| Excitatory coefficient of $Mg^{2+}$ on NCC activity ($\alpha_{NCC,Mg}$) | 0.3 (*) |
| **CD** | |
| $Ca^{2+}$ promoting coefficient for apical HATPase activity of type A OMCD cells ($\alpha_{HATP,Ca}$) | 2 [27] |
| $Mg^{2+}$ promoting coefficient for apical HATPase activity of type A OMCD cells ($\alpha_{HATP,Mg}$) | 1.2 (*) |
| $Ca^{2+}$ inhibitory coefficient for apical water permeability of IMCD cells ($\alpha_{Pf,Ca}$) | $-3/8$ [27] |
| $Mg^{2+}$ inhibitory coefficient for apical water permeability of IMCD cells ($\alpha_{Pf,Mg}$) | $-0.225$ (*) |

## 3. Results

### 3.1. Baseline Results

Using rat parameters, we computed the model's luminal fluid flow, luminal fluid solute concentrations, cytosolic solute concentrations, membrane potential, and fluxes. Figure 2 shows the predicted segmental delivery, transport, and luminal fluid concentration of $Mg^{2+}$ and $Ca^{2+}$ along the nephron. Results for other electrolytes and water can be found in Ref. [32].

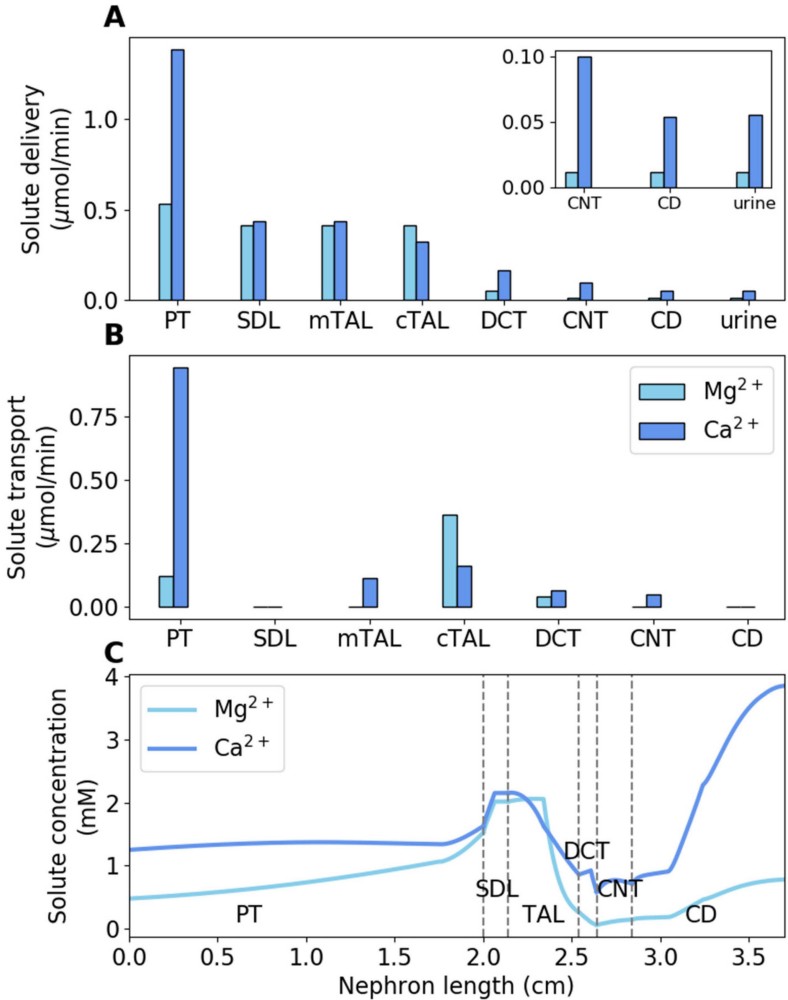

**Figure 2. Baseline results.** (**A**) Delivery of $Mg^{2+}$ and $Ca^{2+}$ to key nephron segments in male rats, given per kidney. (**B**) $Mg^{2+}$ and $Ca^{2+}$ transepithelial transport along key nephron segments in male rats, given per kidney. (**C**) Luminal $Mg^{2+}$ and $Ca^{2+}$ concentrations along key nephron segments in male rats. PT, proximal tubule; SDL, short descending limb; mTAL, medullary thick ascending limb; limb; cTAL, cortical thick ascending limb; DCT, distal convoluted tubule; CNT, connecting tubule; CD, collecting duct.

The majority of the filtered $Ca^{2+}$ is reabsorbed along the proximal tubule, accounting for 68% of the filtered load; by contrast, $Mg^{2+}$ reabsorption along this segment accounts for only 22% of the filtered load. Since $Mg^{2+}$ reabsorption is low along the proximal tubule, $Mg^{2+}$ concentration increases by ~3-fold [4]. The majority of the overall $Mg^{2+}$ transport occurs downstream along the cortical thick ascending limb (68% of the filtered load), where the lumen-positive membrane potential drives $Mg^{2+}$ reabsorption via the paracellular pathway. The fractional reabsorption of $Ca^{2+}$ along the medullary and cortical thick ascending limb is 20%. The final nephron segment that transports $Mg^{2+}$ is the distal convoluted tubule, where 6.6% of the filtered $Mg^{2+}$ is reabsorbed. The fractional reabsorption of $Ca^{2+}$ along the distal convoluted tubule and connecting tubule is 7.9%. Finally, fractional urinary $Mg^{2+}$ and $Ca^{2+}$ excretions are 3.2% and 3.9%, respectively.

### 3.2. Effect of Loop Diuretics

Loop diuretics inhibit NKCC2, which is expressed on the apical membrane of the thick ascending limb. We simulated the effect of acute administration of loop diuretics by inhibiting NKCC2 activity by 70%. We assumed that the NKCC2 inhibitor was administered for long enough to significantly impair the kidney's ability to generate an axial osmolality gradient. The cortical interstitial concentrations were assumed to remain unchanged. Since the concentrating mechanism of the outer medulla is significantly impaired following complete NKCC2 inhibition, the interstitial concentrations of $Mg^{2+}$ and $Ca^{2+}$ at the outer-inner medullary boundary are lowered to 0.77 mM (from a baseline value of 0.96 mM) and 2.0 mM (from a baseline value of 2.5 mM), respectively. At the papillary tip, the interstitial concentrations of $Mg^{2+}$ and $Ca^{2+}$ are reduced to 1.0 mM (from 1.54 mM) and 2.62 mM (from 4.0 mM), respectively. For changes in the interstitial concentrations of $Na^+$, $K^+$, $Cl^-$, and urea, refer to [9].

The predicted $Mg^{2+}$ and $Ca^{2+}$ transport along the thick ascending limb and distal tubules and urinary $Mg^{2+}$ and $Ca^{2+}$ excretions following NKCC2 inhibition in male rats are shown in Figure 3. Our NKCC2 inhibition simulations predicted the fractional $Mg^{2+}$ reabsorption along the cortical thick ascending limb to decrease to 57% from the baseline fractional reabsorption of 69% (Figure 3). Administration of furosemide, a loop diuretic, to male mice increased TRPM6 mRNA expression by 30% [33]. Our model simulations predicted a 68% increase in TRPM6/7 channel activity to account for the 240% increase in $Mg^{2+}$ excretion in male rats undergoing furosemide treatment [34]. Fractional $Ca^{2+}$ reabsorption along the thick ascending limb decreased by 17% following NKCC2 inhibition (Figure 3). This resulted in urinary $Ca^{2+}$ excretion increasing to 236% of the baseline excretion value (Figure 3).

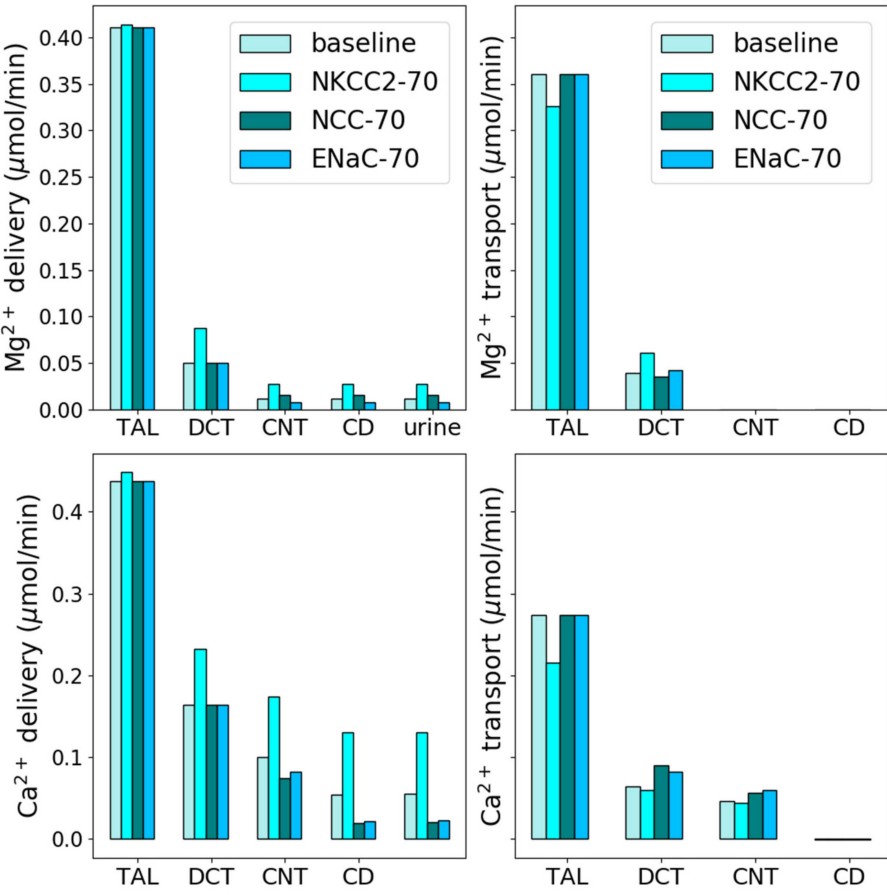

**Figure 3. Effects of diuretic treatment.** Delivery and transport of $Mg^{2+}$ and $Ca^{2+}$ along key nephron segments in male rats under normal conditions and 70% inhibition of NKCC2, NCC, and ENaC. The values are given per kidney. TAL, thick ascending limb; limb; DCT, distal convoluted tubule; CNT, connecting tubule; CD, collecting duct.

### 3.3. Thiazide Diuretics

Thiazide diuretics inhibit NCC, which is expressed along the apical membrane of the distal convoluted tubule. We simulated the effect of acute administration of thiazide diuretics by inhibiting NCC activity by 70%. In the NCC inhibition simulations, baseline interstitial concentration profiles were used.

The predicted fractional $Mg^{2+}$ excretion after NCC inhibition increased to 3.1% from the baseline value of 2.8% (Figure 3). This is in agreement with experimental data where male rats treated with bendrofluazide, a thiazide diuretic, did not show any significant change in $Mg^{2+}$ excretion [35]. Acute administration of chlorothiazide to male mice increased TRPV5 mRNA expression by 40–80% and decreased $Ca^{2+}$ excretion by ~60% [36]. Accordingly, we increased TRPV5 activity by 52% following NCC inhibition in our model. This decreased the predicted $Ca^{2+}$ excretion by 63% from the baseline excretion value (Figure 3).

### 3.4. K-Sparing Diuretics

K-sparing diuretics, such as amiloride, block $Na^+$ uptake through ENaC, expressed on the apical membrane of the late distal convoluted tubule as well as along the full length of the connecting tubule and collecting ducts. In our model, we simulated the effect of K-sparing diuretics by reducing ENaC activity by 70%.

ENaC inhibition hyperpolarizes the luminal membrane potential and increases $K^+$, $Ca^{2+}$, and $Mg^{2+}$ uptake [37–40]. Our model simulations predicted $Mg^{2+}$ reabsorption along the distal convoluted tubule to increase by 8.2% and $Ca^{2+}$ reabsorption along the distal convoluted tubule and connecting tubule to increase by 29% (Figure 3). These increased reabsorptions decreased $Mg^{2+}$ and $Ca^{2+}$ excretions by 31% and 56%, respectively (Figure 3).

### 3.5. TRPM6/7 Inhibition

Kidney-specific TRPM6 [41] and TRPM7 [42] knock-out mice did not display hypomagnesemia and increased urinary $Mg^{2+}$ excretion. This indicates that in mice, there must be other $Mg^{2+}$ uptake mechanisms in the distal convoluted tubule. In fact, Verschuren et al. [43] reported that fluid shear stress (FSS) stimulated $Mg^{2+}$ uptake in mDCT15 cells, and this uptake was independent of the TRPM6 and TRPM7 channels. However, the pathways or regulatory mechanisms for this FSS-sensitive $Mg^{2+}$ uptake are unclear and hence not included in our present model.

To simulate TRPM6/7 knock-out experiments, we inhibited the TRPM6/7 channel by 100%. Our model predicted fractional $Mg^{2+}$ excretion to increase to 8.8% (from baseline 3.2%), which is significantly above the 2–5% physiological fractional $Mg^{2+}$ excretion. How much should the $Mg^{2+}$ uptake through the FSS-sensitive $Mg^{2+}$ channel be for fractional $Mg^{2+}$ excretion to be within the physiological range when TRPM6/7 is completely inhibited? Our simulations predicted that if $Mg^{2+}$ uptake through the FSS-sensitive channel is at least 60% of the $Mg^{2+}$ uptake through TRPM6/7, then the fractional $Mg^{2+}$ excretion becomes 4.3% (within the physiological range).

## 4. Discussion

Calcium ($Ca^{2+}$) and magnesium ($Mg^{2+}$) are both essential for cellular function. The homeostasis of these cations must be tightly regulated, and that balance is facilitated by intestinal absorption and renal excretion. For $Na^+$, $Cl^-$, $K^+$, $Ca^{2+}$, and many other major filtered solutes, most of the renal reabsorption occurs along the proximal convoluted tubule (about 1/2 to 2/3 in rats); the same is true for water. Thus, the luminal concentrations of these solutes, including $Ca^{2+}$, remain close to plasma along the proximal convoluted tubule. The majority of proximal tubule $Ca^{2+}$ reabsorption occurs via a passive paracellular process, driven by $Na^+$ reabsorption mediated primarily by NHE3 and subsequent water reabsorption. In contrast, only 15–25% of the filtered $Mg^{2+}$ load is reabsorbed along the proximal tubule. As a result, its concentration rises significantly along the proximal tubule. For $Mg^{2+}$, most of the reabsorption occurs along the cortical thick ascending limb (about 60–70%), while somewhat unexpectedly, essentially none occurs along the medullary

thick ascending limb. Most of the remainder of the $Mg^{2+}$ is reabsorbed along the distal convoluted tubule.

What difference does it make for the cortical thick ascending limb and distal convoluted tubule to handle most of the $Mg^{2+}$ transport instead of the proximal tubule, as in the case of $Na^+$ and $Cl^-$? Having these distal segments responsible for transporting a substantial fraction of the filtered $Mg^{2+}$ load via the pathways that can be regulated may give the kidney a better ability to regulate $Mg^{2+}$ balance. Recall that the plasma $Mg^{2+}$ level is orders of magnitude lower than $Na^+$ or $Cl^-$. Thus, to maintain plasma $[Mg^{2+}]$ within a narrow range, the ability to fine-tune renal $Mg^{2+}$ transport is particularly crucial. Parathyroid hormone, for instance, increases $Mg^{2+}$ reabsorption in both the cortical thick ascending limb and distal convoluted tubule [44]. Transport of $Mg^{2+}$ along these segments can also be regulated by hormones such as calcitonin, vasopressin, glucagon, and $\beta$-adrenergic agonists [45]. Coincidentally, some common diuretics also target these segments.

The goal of this study is to better understand the impact of diuretics on renal $Ca^{2+}$ and $Mg^{2+}$ transport. Diuretics are medications that reduce fluid buildup in the body and are often employed in the management of hypertension, edema, and various other conditions influenced by changes in electrolyte transport. In the context of kidney function, diuretics often focus on transport proteins or mechanisms vital for the reabsorption of $Na^+$, $Cl^-$, and water. Considering that renal $Ca^{2+}$ and $Mg^{2+}$ transports are driven primarily by the electrochemical gradients established by tubular NaCl transport processes, potential modifications in the renal handling of $Ca^{2+}$ and $Mg^{2+}$ by the administration of diuretics deserve scrutiny.

Loop diuretics, such as furosemide, induce notable natriuresis by targeting the thick ascending limb of the nephron; specifically, they inhibit NKCC2-dependent transport by competing for the chloride ($Cl^-$) binding site [46]. Given that the inhibition of NKCC2 reduces the lumen-positive transepithelial voltage gradient across the thick ascending limb epithelium [47], it is unsurprising that the transport of $Mg^{2+}$ and $Ca^{2+}$ in this segment is substantially decreased [48,49]. In fact, loop diuretics like furosemide induce hypercalciuria and hypermagnesuria in both experimental animals [50] and human subjects [51]. Thiazide diuretics induce a natriuretic response by inhibiting NCC and blocking NaCl transport in the distal convoluted tubule. A hypocalciuric effect has been reported following thiazide treatment [52]. Consistent with the drug's mode of action, among hypertensive individuals undergoing chronic thiazide treatment, there is a slight decrease in serum $Mg^{2+}$ levels compared to those not taking diuretics [53], although the effect appears to be subtle. K-sparing diuretics function by inhibiting ENaC, a protein expressed in the late distal convoluted tubule, connecting tubule, and collecting duct of the kidney. Research has confirmed that K-sparing diuretics impact the urinary excretion of $Ca^{2+}$ and $Mg^{2+}$ in both human subjects and animals. Hypertensive individuals undergoing K-sparing diuretics treatment exhibit reduced urinary $Ca^{2+}$ excretion [53] and elevated serum $Mg^{2+}$ levels compared to those not receiving treatment [53]. The effect of these three classes of diuretics on urinary $Mg^{2+}$ and $Ca^{2+}$ excretions has been summarized in Figure 4.

The present study considers renal $Ca^{2+}$ and $Mg^{2+}$ transport under normal physiological conditions. The homeostasis of these electrolytes is altered during pregnancy, lactation, and dietary restriction, as well as in diseases such as diabetes and chronic kidney disease. To utilize the present model for in silico studies of how the kidney adapts in terms of $Ca^{2+}$ and $Mg^{2+}$ transport under these physiological and pathophysiological conditions, one can combine the model with computational models of kidney function for a pregnant rat [54,55], a diabetic rat [56–58], and a nephrectomized rat [8,59]. The resulting models may provide insights into altered renal $Ca^{2+}$ and $Mg^{2+}$ transport under these conditions. How do changes in renal $Ca^{2+}$ and $Mg^{2+}$ transport impact whole-body $Ca^{2+}$ and $Mg^{2+}$ homeostasis? A whole-body $Ca^{2+}$ and $Mg^{2+}$ balance model may help answer that question. By incorporating the present model into whole-body electrolyte balance models (e.g., [60–63]), one can obtain an integrative model to study whole-body $Ca^{2+}$ and $Mg^{2+}$ balance.

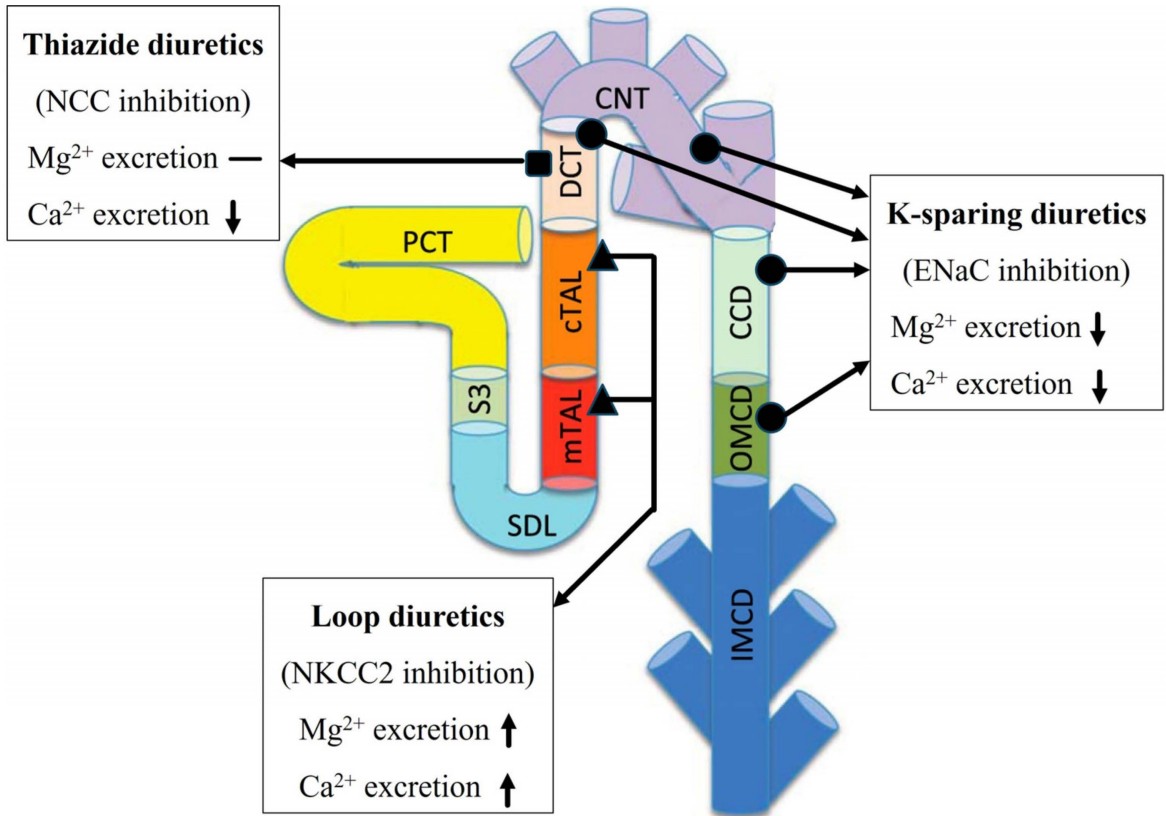

**Figure 4. Summary of the effects of diuretic treatment.** The effect of loop diuretics, thiazide diuretics, and K-sparing diuretics on $Mg^{2+}$ and $Ca^{2+}$ excretion The upward arrow indicates an increase, the downward arrow indicates a decrease, and the dash indicates no significant change. Notations are analogous to Figure 1.

The present model simulates electrolyte and water transport along a superficial nephron, which constitutes only 2/3 of the total nephron population in a rat kidney. The juxtamedullary nephrons, whose loops of Henle extend into various depths of the inner medulla, comprise the remaining nephron population. These two types of nephrons differ in the single-nephron glomerular filtration rate, transport area, and transporter activities. This study focuses on a superficial nephron model to gain a clearer understanding of segmental $Mg^{2+}$ transport. In future studies, developing a kidney model that incorporates both types of nephrons will enable more accurate predictions of urinary excretion rates.

Another limitation of this model is that it does not include cyclin and CBS domain divalent metal cation transport mediator 2 (CNNM2), expressed on the basolateral membrane, which is also potentially involved in $Mg^{2+}$ transport along the distal convoluted tubule. CNNM2 has been shown to mediate both cellular $Mg^{2+}$ influx and efflux. The role of CNNM2 as an $Mg^{2+}$ transporter has been openly debated (in favor [64]; opposing view [65]). A common view is that CNNM2 is not a $Mg^{2+}$ transporter by itself but is an important protein for regulating $Mg^{2+}$ homeostasis [65,66]. Due to these conflicting views on the role of CNNM2 as a $Mg^{2+}$ transporter, we did not include it in our present model. CNNM2 can be incorporated into our model in the future when more consistent experimental outcomes are available.

**Author Contributions:** Conceptualization, A.T.L.; Methodology, P.D. and A.T.L.; Software, Validation, Formal Analysis, and Investigation, P.D.; Resources, A.T.L.; Data Curation, P.D.; Writing—Original Draft, P.D. and A.T.L.; Writing—Review and Editing, P.D. and A.T.L.; Visualization, P.D.; Supervision, P.D. and A.T.L.; Funding Acquisition, A.T.L. All authors have read and agreed to the published version of the manuscript.

**Funding:** This research was funded by the Canada 150 Research Chair program, the National Sciences and Engineering Research Council of Canada (NSERC) Discovery Grant (Grant number: RGPIN-2019-03916), and the Canada Institutes of Health Research (CIHR) Project Grant (Grant number: TNC-174963) (to A.T.L.).

**Data Availability Statement:** The code used for this study can be accessed at https://github.com/Pritha17/Nephron-Mg_Ca_transport (16 October 2023).

**Conflicts of Interest:** The authors declare no conflicts of interest.

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
