# Peer review of "Magnesium and Calcium Transport along the Male Rat Kidney: Effect of Diuretics"

_mca, doi:10.3390/mca29010013_

Round 1

Reviewer 1 Report

Comments and Suggestions for Authors

Dutta and Layton applied an in silico simulation of the rat nephron to anticipate the impact of diuretics on the renal regulation of magnesium and calcium. The renal reabsorption of these cations is a highly complicated process, characterized by distinct organization across various nephron segments. Consequently, a reliable model of these mechanisms holds the potential to greatly enhance our understanding of the renal excretion of magnesium and calcium, as well as the repercussions of medications on kidney function in both healthy and diseased states. Nevertheless, this study prompts concerns regarding the overall relevance of the proposed model. As a result, it is advisable to undertake substantial revisions to the suggested model, as elaborated below.

Any theoretical model critically depends on the accuracy of the input parameters. The authors have postulated the pivotal role of transcellular magnesium transport through the DCT segment in determining the final excretion fraction of this cation. This idea has been extensively discussed in numerous review articles over several decades. Regrettably, such didactic efforts are often perceived as 'textbook knowledge' used to support other theories, such as the explanation of the impact of gene mutations or pharmacological agents on the systemic balance of magnesium. The authors should reference original studies that precisely demonstrate how a DCT-restricted deletion of transcellular magnesium transport affects the ultimate urinary magnesium concentrations. To my knowledge, direct functional evidence (not RNA sequencing or antibody-based tissue section analysis!) specifically addressing this matter remains unavailable.

Furthermore, the proposed role of TRPM6 in the DCT function seemingly overlooks experimental data. In humans, patients with TRPM6 mutations primarily develop systemic magnesium deficiency due to inadequate magnesium absorption in the intestines. This aligns with experimental data derived from mice with a Trpm6 KO mutation. Paradoxically, mice with a kidney-specific Trpm6 KO exhibit normal urinary magnesium excretion rates and unaltered serum magnesium levels, which underscores the notion that intestinal magnesium intake is capable of compensating for the absence of renal TRPM6. Notably, none of the available Trpm6 KO mouse strains have been assessed for changes in transcellular magnesium transport in the DCT. Therefore, it is unlikely that an indirect pharmacological modulation of TRPM6 in the DCT would influence renal magnesium handling and, more crucially, organismal magnesium balance.

Furthermore, the authors have not considered the role of other proteins, such as TRPM7 and CNNM proteins, in their analysis.

Author Response

Thank you very much for taking the time to review this manuscript. Please find the detailed responses below and the corresponding revisions in track changes in the re-submitted manuscript.

Comment 1. Any theoretical model critically depends on the accuracy of the input parameters. The authors have postulated the pivotal role of transcellular magnesium transport through the DCT segment in determining the final excretion fraction of this cation. This idea has been extensively discussed in numerous review articles over several decades. Regrettably, such didactic efforts are often perceived as 'textbook knowledge' used to support other theories, such as the explanation of the impact of gene mutations or pharmacological agents on the systemic balance of magnesium. The authors should reference original studies that precisely demonstrate how a DCT-restricted deletion of transcellular magnesium transport affects the ultimate urinary magnesium concentrations. To my knowledge, direct functional evidence (not RNA sequencing or antibody-based tissue section analysis!) specifically addressing this matter remains unavailable.

Furthermore, the proposed role of TRPM6 in the DCT function seemingly overlooks experimental data. In humans, patients with TRPM6 mutations primarily develop systemic magnesium deficiency due to inadequate magnesium absorption in the intestines. This aligns with experimental data derived from mice with a Trpm6 KO mutation. Paradoxically, mice with a kidney-specific Trpm6 KO exhibit normal urinary magnesium excretion rates and unaltered serum magnesium levels, which underscores the notion that intestinal magnesium intake is capable of compensating for the absence of renal TRPM6. Notably, none of the available Trpm6 KO mouse strains have been assessed for changes in transcellular magnesium transport in the DCT. Therefore, it is unlikely that an indirect pharmacological modulation of TRPM6 in the DCT would influence renal magnesium handling and, more crucially, organismal magnesium balance.

Response 1: We thank the reviewer for the comments. The effect of TRPM6 inhibition on urinary Mg2+ excretion is discrepant. Though some studies show that TRPM6 inhibition does not alter urinary Mg2+ excretion (doi: 10.7554/eLife.20914), there are other studies that demonstrate that TRPM6 inhibition in the kidney impairs renal Mg2+ reabsorption and causes increased Mg2+ excretion (doi: 10.1038/s41467-021-24063-2, doi: 10.1038/ng889, doi: 10.1038/ng901). To evaluate the effect of TRPM6 on urinary Mg2+ excretion, we conducted TRPM6 inhibition simulations with our model, where we inhibited TRPM6 by 70% and 100%. The results of these simulations have been included in a new section 3.5 TRPM6 inhibition in the revised manuscript. We have also included a discussion of the discrepant experimental studies on the effect of TRPM6 inhibition on urinary Mg2+ excretion in this section (section 3.5).

Comment 2. Furthermore, the authors have not considered the role of other proteins, such as TRPM7 and CNNM proteins, in their analysis.

Response 2: We thank the reviewer for bringing up this important point. We acknowledge that TRPM7 and CNNM proteins play an important role in renal Mg2+ transport. However, studies on the function of TRPM7 and CNNM2 as Mg2+ transporters are not consistent. For this reason, we did not include TRPM7 and CNNM2 in our present model. This is a limitation of our model which we have discussed in the Discussion section of the revised manuscript. We can incorporate these two Mg2+ channels in our model in future when more consistent experimental studies are available.

Reviewer 2 Report

Comments and Suggestions for Authors

This work is exciting and clearly written.

1. However, I would request the authors to include a figure as a schematic diagram in the discussion part as a summary in order to get a better understanding. 
2. The limitations of this study should be mentioned in the work.

Author Response

Thank you very much for taking the time to review this manuscript. Please find the detailed responses below and the corresponding revisions in track changes in the re-submitted manuscript.

This work is exciting and clearly written.

Comment 1. However, I would request the authors to include a figure as a schematic diagram in the discussion part as a summary in order to get a better understanding.

Response 1: We thank the reviewer for the suggestion. We have included a schematic diagram of the main findings (Figure 4) in the Discussion section of the revised manuscript.

Comment 2. The limitations of this study should be mentioned in the work.

Response 2: We have included the limitations of our study in the Discussion section of the revised manuscript.

Round 2

Reviewer 1 Report

Comments and Suggestions for Authors

Comment 1. Any theoretical model critically depends on the accuracy of the input parameters. The authors have postulated the pivotal role of transcellular magnesium transport through the DCT segment in determining the final excretion fraction of this cation. This idea has been extensively discussed in numerous review articles over several decades. Regrettably, such didactic efforts are often perceived as 'textbook knowledge' used to support other theories, such as the explanation of the impact of gene mutations or pharmacological agents on the systemic balance of magnesium. The authors should reference original studies that precisely demonstrate how a DCT-restricted deletion of transcellular magnesium transport affects the ultimate urinary magnesium concentrations. To my knowledge, direct functional evidence (not RNA sequencing or antibody-based tissue section analysis!) specifically addressing this matter remains unavailable.

Furthermore, the proposed role of TRPM6 in the DCT function seemingly overlooks experimental data. In humans, patients with TRPM6 mutations primarily develop systemic magnesium deficiency due to inadequate magnesium absorption in the intestines. This aligns with experimental data derived from mice with a Trpm6 KO mutation. Paradoxically, mice with a kidney-specific Trpm6 KO exhibit normal urinary magnesium excretion rates and unaltered serum magnesium levels, which underscores the notion that intestinal magnesium intake is capable of compensating for the absence of renal TRPM6. Notably, none of the available Trpm6 KO mouse strains have been assessed for changes in transcellular magnesium transport in the DCT. Therefore, it is unlikely that an indirect pharmacological modulation of TRPM6 in the DCT would influence renal magnesium handling and, more crucially, organismal magnesium balance.

Response 1: We thank the reviewer for the comments. The effect of TRPM6 inhibition on urinary Mg2+ excretion is discrepant. Though some studies show that TRPM6 inhibition does not alter urinary Mg2+ excretion (doi: 10.7554/eLife.20914), there are other studies that demonstrate that TRPM6 inhibition in the kidney impairs renal Mg2+ reabsorption and causes increased Mg2+ excretion (doi: 10.1038/s41467-021-24063-2, doi: 10.1038/ng889, doi: 10.1038/ng901). To evaluate the effect of TRPM6 on urinary Mg2+ excretion, we conducted TRPM6 inhibition simulations with our model, where we inhibited TRPM6 by 70% and 100%. The results of these simulations have been included in a new section 3.5 TRPM6 inhibition in the revised manuscript. We have also included a discussion of the discrepant experimental studies on the effect of TRPM6 inhibition on urinary Mg2+ excretion in this section (section 3.5).

New comment 1R

It is appreciated that the authors attempt to evaluate the original studies instead of referring to review articles. Unfortunately, the selected three references are irrelevant to addressing my concerns, as explained below.

The DCT-centric theory claims that transcellular magnesium transport through this nephron segment determines the final urinary magnesium content. All three mentioned studies (doi: 10.1038/s41467-021-24063-2, doi: 10.1038/ng889, doi: 10.1038/ng901) do NOT investigate the transcellular magnesium transport through the DCT and do NOT demonstrate impact of this mechanism on renal excretion rate of magnesium, reinforcing my original request: The authors should reference original studies that precisely demonstrate how a DCT-restricted deletion of transcellular magnesium transport affects the ultimate urinary magnesium concentrations.

My second concern is the lack of functional data supporting the popular idea that TRPM6 controls the transcellular magnesium transport through the DCT. All three studies do NOT investigate the impact of TRPM6 on this cellular process in DCT but only discuss this scenario.

My third concern is that the authors should be more accurate in interpreting data reported for HSH patients and mice with the deletion of Trpm6.  Thus, Walder et al.  (doi: 10.1038/ng901) did not claim that hypomagnesemia in HSH patients is solely caused by abnormal magnesium handling in the kidney but instead only suggested that TRPM6 may operate both in the kidney and intestine. Please also note that Figure 4 of Walder et al. paper reported urinary excretion of magnesium in individuals with TRPM6 mutation compared to data for healthy individuals published elsewhere. This approach has several pitfalls. HSH patients were subjected to life-long magnesium supplementation before the test, implying that a similar treatment should be conducted with the control group for a reliable comparison. In addition, a control group should match the affected group regarding gender, age, and lifestyle, which is not the case for the data shown in Figure 4. Consequently, Walder et al. raised a rather general careful note that abnormal magnesium handling by the kidney may occur in HSH patients.  Along these lines, it is unclear why the authors refer to this study as proof of the TRPM6-DCT theory.

Similarly, Schlingmann et al. (doi: 10.1038/ng889) did show (and do not claim!) the primary role of the DCT function in systemic magnesium balance. The authors reported in Table 1 fractional magnesium excretion rates in the patients with TRPM6 mutations without a proper control group (matched by age, gender, and magnesium supplementation). Hence, it is unclear how this approach could confirm that TRPM6 KO halts the DCT magnesium transport and exactly this mechanism causes systemic magnesium deficiency.

Funato et al. (doi: 10.1038/s41467-021-24063-2) used Six2-Cre for conditional inactivation of Trpm6 in mouse kidneys. Funato et al. reported a modest reduction of magnesium in serum (~20%) in Trpm6 gene-deficient mice without a remarkable impact on the growth and survival of the mutant mice. Ironically, this finding strongly supports the idea that the lack of TRPM6 in the intestine underlines drastic hypomagnesemia incompatible with the life of humans or mice with a global TRPM6 KO.  Moreover, the study of Funato et al. contains numerous technical pitfalls. For instance, the authors compared the phenotype of Trpm6+/+;Six2-Cre and Trpm6fl/fl;Six2-Cre. In such settings, the possible impact of the floxed sequence in the Trpm6 locus on the expression of TRPM6 protein in the whole body cannot be ruled out and should be addressed experimentally. A common and straightforward approach to avoid this problem is to use Trpm6fl/fl littermates instead. The magnesium urinary excretion rate should be normalized to very different body sizes of the female and male mice examined, but this procedure was skipped in the present study. Notably, the authors did not study magnesium transport in the DCT and the impact of the Trpm6 mutation on this transport mechanism.

The authors should specify why the study, doi: 10.7554/eLife.20914 should be neglected.  

Finally, the author should refer to original publications supporting a new assumption that diuretics can inhibit TRPM6 by 70% and 100% and why this inhibition should specifically affect TRPM6 but not TRPM7. Why will TRPM6 (TRPM7) be suppressed selectively in the kidney but not in the intestine?  

Comment 2. Furthermore, the authors have not considered the role of other proteins, such as TRPM7 and CNNM proteins, in their analysis.

Response 2: We thank the reviewer for bringing up this important point. We acknowledge that TRPM7 and CNNM proteins play an important role in renal Mg2+ transport. However, studies on the function of TRPM7 and CNNM2 as Mg2+ transporters are not consistent. For this reason, we did not include TRPM7 and CNNM2 in our present model. This is a limitation of our model which we have discussed in the Discussion section of the revised manuscript. We can incorporate these two Mg2+ channels in our model in future when more consistent experimental studies are available.

New comment 2R

The authors need to justify the statement that the function of TRPM7 and CNNM2 as Mg2+ transporters is not consistent. In my view, many independent studies clearly show it.                                                                                                                                       

Round 3

Reviewer 1 Report

Comments and Suggestions for Authors

The authors have addressed most of my concerns, and the latest version of the manuscript now appears more balanced. I have only one final comment for the authors, which they may wish to take into account in their subsequent publications. The authors highlighted a study doi:10.1074/jbc.M311201200 showing a role for TRPM6 in transcellular magnesium transport through the DCT. In fact, this manuscript focuses on patch-clamp analysis of TRPM6 overexpressed in HEK293 cells and immunostaining of kidney and intestinal tissue sections. Therefore, these experiments do not allow us to conclude that TRPM6 mediates transcellular magnesium transport through the DCT.